# Relevant Skills for Employment and Entrepreneurship in the Agri-Food Sector

Dan Bodescu [ID], Alina Sîrghea, Roxana Nicoleta Raţu [ID], Ciprian Chiruţă [ID], Răzvan-Nicolae Mălăncuş [ID], Dan Donosă and Alexandru-Dragoș Robu *

Faculty of Agriculture, "Ion Ionescu de la Brad" Iași University of Life Sciences, M. Sadoveanu Alley, No. 3, 700490 Iasi, Romania; dbodescu@uaiasi.ro (D.B.); asirghea@uaiasi.ro (A.S.); roxana.ratu@uaiasi.ro (R.N.R.); kyru@uaiasi.ro (C.C.); razvanmalancus@uaiasi.ro (R.-N.M.); donosad@uaiasi.ro (D.D.)
* Correspondence: drobu@uaiasi.ro; Tel.: +40-740243184

**Abstract:** In the current social, economic, natural and geopolitical context, there is an acute need for research on professional and entrepreneurial skills in line with the challenges and opportunities of the rapidly changing global economy. The aim of this study was to determine the skills required by employees and entrepreneurs in the agri-food sector. This research study consisted of interviews, questionnaires and focus groups on a sample of 111 employers, 288 students and 139 teachers from the North-East Development Region of Romania. The most important skills needed by higher education graduates in the agri-food sector were communication, learning and social skills, with values of 90.5, 74.3 and 70.6 points, respectively. The least appreciated skills were cultural, linguistic and mathematical skills, with values of 17.5, 36.9 and 43.8 points. For developing and running an entrepreneurial activity, the subjects appreciated communication (91.0 points), as in the case of employment-related skills, followed by economic skills (81.4 points) and learning skills (75.4 points). Additional efforts are needed to increase the relevance of practical activities in correlation with the skills required by the market, and it is necessary to strengthen the current partnerships and create new partnerships between universities and the economic environment by employing public and research entities.

**Keywords:** agri-food; higher education; employment; entrepreneurship; communication skills; learning skills; social abilities; economic skills

## 1. Introduction

The human resources department is a crucial component of any organization, as it is responsible for managing the people working within the company. The effectiveness of human resource management can have a significant impact on an organization's competitiveness, as it influences the recruitment, training and retention of talented employees, as well as the development of a positive organizational culture [1].

Numerous studies have highlighted the importance of human resources in improving organizational competitiveness. For example, research has shown that effective HR practices can lead to increased employee satisfaction, increased productivity and improved financial performance [1]. Furthermore, other research has suggested that investment in human resources can contribute to better innovation and greater adaptability in a rapidly changing business environment [2].

Given the increasing share of remote work, effective HR management is needed to build virtual teams and maintain employee engagement [3].

The agri-food sector is an essential component of many national and regional economies, providing food and jobs for millions of people around the world. In 2021, the share of agriculture in world GDP was about 4% and the added value had increased by 84% between 2000 and 2021, up to USD 3.7 trillion. In Europe, the share of agriculture in total GDP was

9.0% and in Romania, it was 4.1%. A total of 873 million people were employed in the agricultural sector in 2021, i.e., 27% of the global workforce. In Europe, 5.1 percent of the employed population had a job in agriculture and in Romania, this figure was 18.6% [4].

In this context, the effective management of human resources is essential for the competitiveness and sustainability of agri-food enterprises [5]. In the current social context, the management of human resources in the agri-food sector is essential for building a diversified and inclusive workforce. This includes providing training and development opportunities to all employees regardless of their background and ensuring fair and equal treatment for all [1].

Jackson, S.E. and Ruderman, M. state that effective human resource management can help agribusinesses attract, develop and retain top talent employees, increasing their competitiveness. This includes offering employees competitive wages and benefits, improving working conditions and creating a positive work culture that encourages employee engagement and innovation [3,6]. Also, an essential role in promoting sustainable activities is played by reducing the environment impact of agribusinesses by providing training of sustainable agricultural practice, promoting waste reduction and energy efficiency and creating a culture of environmental responsibility [7]. By prioritizing human resource management, agribusinesses can build a skilled and engaged workforce, promote sustainable practices and navigate the challenges of a rapidly changing global economy [3].

In today's social, economic, natural and geopolitical context, there is an urgent need for research on professional and entrepreneurial skills to address the challenges and opportunities of the rapidly changing global economy [8]. Professional and entrepreneurial skills are essential for setting up new businesses, promoting innovation and stimulating economic growth. However, there is a lack of research exploring how these skills can be developed and applied in different contexts, including emerging markets and developing countries [9,10].

Research in the field of professional and entrepreneurial skills is driven by the need to develop a highly qualified workforce that can meet the demands of the global economy [11]. This includes promoting the development of entrepreneurship and innovation, which are key drivers of economic growth and job creation [12]. In addition, research on professional and entrepreneurial skills can contribute to promoting social and economic development, reducing poverty and inequality and promoting sustainable development.

Every society has the obligation to provide its young citizens with the necessary education and vital skills to become productive and prosperous members of the community they belong to. However, young people suffer from a skills gap where they do not master the minimum package of education needed to get a job and maintain a decent way of life [13,14]. Today's labor market requires most members to have some form of postgraduate education [15], an aspect also confirmed by the results of the Agree (USA) report, which emphasizes that agricultural education must produce a workforce that is prepared for the growing contemporary challenges in agriculture [16]. The food industry is the largest manufacturing sector in the EU, but labor productivity is lower than in most other industrial sectors, as the percentage of highly qualified staff is low and the transfer of scientific knowledge and innovation to industry is limited (especially in the case of SMEs); therefore, it is difficult to implement new technologies and knowledge [17]. Romania's tertiary education graduation rate is the lowest among all EU countries. In 2021, this rate was 23.3%, almost half of the EU average (41.2%). This phenomenon, leading to a shortage of highly skilled professionals, is compounded by emigration, which further reduces the availability of highly skilled and highly educated workers [18].

In research started by the European Food Studies and Training Alliance, the following categories of skills were noted as important: In the food processing sector, product compliance with the legislation in force is very important (48%), followed by work methods and production optimization (43%). In the food safety sector, food hygiene is very important (77%), as is compliance with the legislation in force (50%). In the field of marketing, ongoing study of the competitive market is marked as important (41%), as are export capacity

(38%) and market investigation capacity (37%). In addition to these, respondents noted the importance of technical skills (63%), followed by practical skills (58%), communication skills (36%), managerial skills (35%) and finally, digital skills (18%). After the COVID-19 pandemic, which caused crisis situations in most fields, the requirement to demonstrate competence in crisis management or effective communication in such situations was noted.

Therefore, the current state of research on the skills needed by employees and entrepreneurs in the agri-food sector focuses on the need for crisis communication, leadership and decision-making skills, which have become vital transversal skills for any higher education graduate [19]. However, another research study [20] also shows that these skills are not enough, as they involve an interdependent relationship with evidence of meeting deadlines, trust, flexibility and a strong work ethic. Although employers prefer graduates to come equipped with these transferable skills, many of them have not demonstrated these skills in the workplace, creating a skills gap between employer expectations and employee competencies [19].

Jankelová, N. and collab.appreciate that a minimum necessary collection of skills requested by employers in the field of agriculture, which has similarities with the eight key skills established by the Council of Europe, would be composed of [20] the following: academic knowledge in the field; basic reading and writing skills; problem solving/critical thinking; working efficiently under stress; independent work; positive attitude; active listening. We observe that beyond scientific competence in the field, which is absolutely necessary, to some extent, employers are willing to give new employees time to learn on the job. Things are different if the employee does not demonstrate flexibility, resistance to stress, confidence or work ethic.

The modern digital revolution is driven by the transition from analog to digital electronic technologies. In 1986, 99.2% of the globe's storage capacity was analog, while by 2007, 94% of the world's information storage capacity had been converted to digital [21].

Digital skills increase young people's self-confidence in using technology for daily tasks, education, employment or entertainment. Having access to digital devices is necessary but not sufficient for skill development, as there are other factors that affect students' skill levels. Also, although there have been benefits in using technology in education, this does not necessarily indicate that teachers can successfully integrate it into their lessons [22].

*1.1. The University Education System in Romania: Some Aspects Relevant to the Labor Market*

In Romania, there are several government institutions involved in developing the necessary skills in the agri-food sector, among which are the Ministry of Agriculture and Rural Development (MARD), the Ministry of Education and the Ministry of Labor and Social Solidarity. There are also other institutions and organizations in Romania involved in the development of skills in the agri-food sector, such as Commerce and Industry Chambers, Farmers' Associations as well as various projects financed by the European Union.

The national education system, involved in skill creation and development, is made up of all state and private educational units and institutions of various types, levels and forms of organization of training and education activity.

This system is structured into educational levels, so as to ensure the coherence of education and the development of skills according to the age and individual characteristics of the students. Educational modules focused on developing skills intended for the agri-food sector are included both in traditional national curricula as well as in third-party programs organized by public or private institutions.

Non-tertiary postsecondary education, an important source of skills for the agri-food sector, is divided into postsecondary schools and vocational schools and can only be attended by high school graduates, with or without a baccalaureate diploma.

In pre-university education, students can transfer between educational institutions, branches, profiles and specializations, based on specific rules established by the Regulation on the Organization and Operation of Pre-University Education Units.

Higher education is coordinated by the Ministry of Education and Research, respecting the principles of university autonomy. Higher education, the most important source of skill creation for the agri-food sector, is provided by universities, research institutes, study academies and postgraduate study schools [23]. Higher education institutions establish their admissions methodology according to the general criteria established by the Ministry of Education and Research. Only high school graduates with a baccalaureate diploma can be admitted to higher education [23].

Toma, M. believes that the acquisition of skills by trainees is not the result of their spontaneous transformation but the result of a wide and complex series of inhomogeneous actions, intercorrelated from a methodological and organizational point of view, involving human, material, mechanical and financial resources, which must be organized, planned, coordinated, controlled and evaluated in an operational system and by a managerial performance. All of these actions come together in the educational process, resulting in the accumulation of relevant skills [24].

Thus, the educational process can be defined as a system that brings together a set of elements (actions) that interact with each other as a complex of responsibilities, processes or activities that take place in connection with the transformation of young students and trainees organized, led and carried out by teaching staff, with the help of appropriate teaching methods and within a specific methodology in order to obtain the educational product [24].

Through a careful examination of the researcher status in the social sciences and humanities fields according to official documents, the analysis of training programs in doctoral schools both in Romania and abroad (Germany, Austria, France, England, USA, etc.), the systematic study of specialized literature on the topic of research methodology, the reflective analysis of our own experience as doctoral thesis coordinators, as well as guided by the need for expertise in advanced research, we have concluded that the skills attained through the educational process are professional skills and transversal skills [25].

The responsibility of the educational manager can be analyzed in at least two ways, namely in terms of material responsibility and moral responsibility. While material responsibility can be quantified and measured by the way the educational manager complies with his activities in accordance with the national legislation in the field of education and by the way he adheres to the regulations approved by the Board of Directors regarding the proper conduct of all his activities in the educational institution, for moral responsibility, there are no mechanisms that can be accurately observed and thus quantified. Although moral responsibility is a rather diffuse notion, it plays a very important role and can decisively define the manager's work. Ethics in the educational institution must be known and recognized in all activities [26].

In Romania, the acquisition of relevant skills in the agri-food sector is also achieved at pre-university structures. Pre-university education is an integral part of the national education system, bringing together state, private, confessional, authorized and accredited educational units. It is organized by levels, forms of education and, particularly, streams and profiles, ensuring the necessary conditions for the acquisition of key competencies and for progressive professionalization [27].

Butum, L.C. and collaborators found that at the level of Iași county, in the 2022–2023 school year, there are both state pre-university education institutions and private education institutions. In the first category, there are eight units, each of them having one or more specializations or qualifications in the field of agri-food economy, both related to the form of professional and high school education as well as to the form of post-high school education. On the other hand, there are also two accredited private units which cover specializations in the agri-food economy sector and allow students to obtain a qualification [28].

Thus, continuing professional education and training, abbreviated CPE (also known as professional training for adults) and regulated by Government Ordinance no. 129/2000 on the professional training of adults, is available to students with a minimum age of 16 [29].

The purpose of the research presented in this article is to determine the skills needed by employees and entrepreneurs in the agri-food sector. This study includes the following objectives: (1) determination of the skills required by entrepreneurs in the agri-food sector; (2) determining the skills offered by higher education providers dedicated to the agri-food sector; (3) determining the skills obtained by students in the agri-food field.

Current research looking at the demand and supply of skills for the agri-food sector is mainly focused on identifying the skills needed by the agri-food sector, as well as identifying skill gaps and possible solutions to bridge them. Developing a sustainable and resilient European economy has been the focus of literature reviews and bibliometric analyses on the agricultural sector [30]; in particular, researchers have focused on the need for professionals to have the necessary skills to engage in the transition to sustainable agriculture [31] and on farmers' entrepreneurship skills [32]. In addition, empirical research has been conducted on skill requirements for precision agriculture [33], the personal, communication and leadership qualities sought by leaders in the agricultural and natural resource industries [34] and the skills needed by agronomists to support sustainable agriculture [35].

To determine the most important knowledge, skills and competencies for the food industry workforce, Flynn et al. [36] organized workshops with food companies. Mayor et al. [37] conducted a survey of food industry professionals to determine their training needs and compared their results. In addition, Akyazi et al. [38] developed a database of the current and future skills emerging in Industry 4.0 for various professional profiles in the food industry, while Handayani et al. [39] conducted surveys to identify the green skills needs of professional food industry graduates by assessing their current skills in small and medium enterprises in the Thai food industry.

In contemporary labor market assessments, a variety of ideas are used as the basic unit of analysis. The oldest idea is that of "labour", which has the more general connotation of "production asset" and emphasizes analogies rather than differences between individuals. During the time period when production was predominantly determined by muscle power, the terms labor and labor power were commonly used [40].

In Romania, the National Institute of Statistics conducted a study in 2021 entitled "The workforce in the agri-food sector in 2020" [41], which analyzed the characteristics of the labor force in this sector. The study found that the majority of workers in the agri-food sector are men, and a significant proportion of them have no or minimal qualifications.

Another study was carried out by the Bucharest Academy of Economic Studies in 2019, entitled "Skills demand and supply in the Romanian agri-food sector". The study analyzed the skills needs of the Romanian agri-food sector as well as the current supply of skills in this field. The findings of the study included the need for a more integrated approach to education and vocational training in the agri-food sector, as well as the need to increase the level of digitization and innovation in the sector.

A 2020 report of the Romanian Flour, Baking and Beef Industry Association (ROMPAN) [41] shows that there is a growing need for technical and managerial skills in the Romanian food industry, especially in the areas of quality management, food safety and innovation.

A study conducted in 2019 by the consultancy company PwC [42] showed that the food and beverage industry in Romania faces a skills shortage among employees with higher education, especially in the fields of food technology, quality control and product development.

Another study, conducted by the consultancy company Deloitte in 2018 [43], showed that employers in the food industry find it difficult to find candidates with specific skills in areas such as product development, quality management and process control.

Regarding the offer of Romanian universities in the field of the agri-food industry, there are a variety of higher education institutions that offer bachelor's and master's programs in fields such as food technology, bioengineering, food engineering or quality management in the food industry. Romanian universities have started to develop more practical and

labor market-oriented study programs in an attempt to reduce the gap between what is taught and what students are looking for [44–47].

Generally, research on the demand and supply of skills for the agri-food sector is important to understand the needs and challenges of the sector and to identify solutions to address them. This research study can serve as a basis for the development of education and training programs for the agri-food sector, as well as for the identification of career development opportunities in this field.

The Iași University of Life Sciences "Ion Ionescu de la Brad" in Iași (USV Iași), where this study was carried out, is a higher education institution specialized in agronomy and veterinary medicine, with a national and European scope. Its academic offer is defined by accredited faculties with fields of study with professional impact, quality education and high performance in scientific research.

Within USV Iași, the following forms of university studies are carried out: bachelor's education; master's education; doctoral school; postdoctoral school; continuous training; teaching staff training.

Scientific research represents the second major dimension of the activity carried out by the teaching and research staff of USV Iași and is contracted through grants financed by funds obtained in competitions developed by national financiers.

The scientific research capacity of the University has increased, diversified and strengthened through the RENAR accreditation of the Oenology Laboratory and the accreditation of the Laboratory for the expertise, certification and control of genetically modified organisms, the Plant-Soil Analysis Laboratory and the creation of a Research Institute for Agriculture and the Environment (RIAE) financed by structural funds. A phytotron and a greenhouse operate within the University.

In a study of the whole university, the insertion of USV Iași graduates in the academic year 2020–2021 into the labor market was analyzed [48]. In this study, a total of 1093 graduates from the four faculties were monitored. The first analysis was on whether the graduates are employed, looking for jobs or continuing their studies. A total of 666 young people were already active in the labor market and 427 people were looking for jobs.

An important consideration for the prestige of a university is whether its graduates use the knowledge they were taught at their new job or whether have to obtain new skills along the way. A percentage of 59.45% (396 graduates) were employed in the field in which they studied at university and only 40.45% (270 graduates) chose a job in other fields of activity.

An interesting topic in the study was the degree of independence of USV Iași graduates. It was found that 106 graduates became entrepreneurs, 151 were self-employed and 409 were employed.

Regarding the employability of students distributed among the four faculties of the University of Life Sciences, in 2021, the following situation was reported: the Faculty of Agriculture produced 318 employed graduates, the Faculty of Horticulture had 117 graduates, the Faculty of Animal and Animal Resources Engineering produced 132 employed graduates and the Faculty of Veterinary Medicine had 99 employed graduates.

### 1.2. The Main Characteristics of the North-East Development Region

Romania is divided into eight economic development regions: North-East, South-East, South Muntenia, South-West Oltenia, West, North-West, Center and Bucharest-Ilfov.

Benefiting from a rich historical, cultural and spiritual heritage, the North-East region of Romania [49] harmoniously combines the traditional with the modern and the past with the present, having a huge potential for the development of infrastructure, rural areas, tourism and human resources.

The North-East region is composed of six counties: Bacău, Botoșani, Iași, Neamț, Suceava and Vaslui. These regions are waiting to be discovered and promoted. Two advantages stand out in the area: the costs of living, which are relatively low, and the labor

force, which is highly qualified. Information about the population in the area is taken from the National Institute of Statistics following the 2011 Population Census [50].

The population of the North-East region is divided by counties as follows: Bacău (616,168 inhabitants), Botoșani (412,626 inhabitants), Iași (772,348 inhabitants), Neamț (470,766 inhabitants), Suceava (634,810 inhabitants) and Vaslui (395,499 inhabitants).

After Romania's integration into the European Union in 2007, all economic departments began to compare their local evolution with other European countries. A comparison of the evolution of tourism and its degree of development in the North-East region of our country with that of countries such as Poland and Slovakia has aroused the interest of researchers [51].

Due to the strong differences observed in the development regions of Romania between rural areas, which comprise 90% of the Romanian territory and 34% of the entire population, and urban areas, the resources and alternatives that can be taken into account have been evaluated so that the North-East region of Romania can capitalize on them and develop successfully [52].

However, the North-East region remains in the last place in the ranking of EU regions. Research has been conducted on the social and economic dimension of this region's development in order to identify the main problems and perspectives of the region [53].

The research presented in this article is an integral part of a project funded by the Romanian Ministry of Education, grant number CNFIS-FDI-2022-0112/01.04.2022, with the title "Correlation of the educational offer of the University of Life Sciences, Iasi with the application the labor market, counseling and career guidance" (ExcelentProf.). The objectives of the project are as follows: O1. analysis of the insertion of USV Iași graduates into the labor market and their professional careers; O2. analysis of the education plans and skills developed at USV Iași in accordance with labor market demand; O3. consolidation of partnerships between USV Iași and public and private economic organizations. The research presented is part of the analysis of the correlation between the education plans and the skills obtained by graduates and the requirements of the labor market within the first objective stated above.

The novel element proposed by the research presented in this article consists in carrying out a representative survey of how the key skills system agreed upon by the European Union is achieved in our research area. The presented results were generated together with the most important actors in the agri-food labor market of the NE Development Region in Romania. The research design can be replicated in other sectors of activity and can be used by narrower branches of activity to obtain specific information.

## 2. Materials and Methods

The research approach was carried using a mixed structure consisting of two qualitative research methods (interview and focus group) and a quantitative method, namely a questionnaire-based survey. A mixed-methods research design is an approach that combines both qualitative and quantitative methods to gain a deeper understanding of whether the researched phenomenon [54] is comprehensive [55], confirms and enriches the results, captures their complexity [56] and allows theories to be developed and tested [57].

### 2.1. Qualitative Research on Skills Relevant to Employment and Entrepreneurship in the Agri-Food Sector

The first qualitative survey consisted of conducting a semi-structured interview to determine the opinions of employers, teachers and students regarding the need for skills in the agri-food sector.

The design of the research followed the following main stages: objectives, sample and methods of achieving this approach.

The interview had the following objectives: (1) identifying the skills necessary for higher education graduates to be employed in the agri-food sector; (2) prioritizing the skills necessary for higher education graduates to be employed in the agri-food sector; (3) estab-

lishing the main differences between the skills needed by higher education graduates to be employed and the skills needed by higher education graduates to become entrepreneurs in the agri-food sector.

These objectives would help us determine a set of skills that would be the basis for the development of the questionnaire in the next stage [58].

We were motivated to use a semi-structured interview due to the advantages of this research tool, the interest of the subjects in participating, which was correlated with the research objectives, and due to the possibility of obtaining extra information about the subjects' behavior outside the answers given [59].

The number of interview participants was 36 and they were selected following a cluster sampling process, comprising 12 employers, 8 teachers and 16 students. Employers were managers of economic units in the agri-food sector. Teachers were part of the academic community at IULS Iași. Students were selected from each faculty within the University and were in their third or fourth year of studies. These specific subjects were chosen for the following reasons: managers have all the necessary information and are interested in improving the skills of graduates, teachers want to know the skill set needed in the labor market and students are concerned about their professional success.

Data collection was carried out by 6 specialized researchers in teams of two and the research area was represented by the North-East Development Region of Romania.

In order to ensure a better response availability of the employers, they were interviewed at their respective companies. Students and teachers were interviewed at the University.

The interview was carried out through direct individual interaction with the subjects because in this way, relevant information regarding the objectives of our research could be obtained.

To support the specialists in field research, an interview guide was developed, which included guiding questions on interview objectives and explanations on how, when and where to carry out this activity.

The subjects were informed at the beginning of the interviews about the answers that would be requested and how these would be used. It was also explained to them that these would be published without naming their personal data. Accordingly, they were asked for their consent regarding the use of this information.

Subjects' responses were noted at the time of the interview in a synthetic manner and completed at the end of the interview day in a debriefing session. The reports thus obtained were the basis of the interview analysis.

Data analysis was carried out by reducing the data, creating a matrix and drawing conclusions based on the information obtained in the field [60].

All the researchers participated in the data analysis, as some conclusions are usually generated at the field stage and then discussed among the research team [61].

The period in which the interviews were carried out was the second quarter of 2023. This period of time was chosen because the managers of the agricultural units are less busy during this period, as it is between crop establishment in the spring and crop harvest in the fall. The duration of the individual interviews was approx. 1 h each.

The third stage of the research consisted of conducting a two-way focus group session, which involved collecting data from a small group of participants who are relevant to the research objectives and can provide opinions on its topic.

A focus group was used in this research due to the advantages that this method presents, such as facilitating the exchange of opinions, the possibility of obtaining relevant information and an interactive approach that can reveal deep perceptions and needs [62]. The research design followed the following stages: objectives, establishment of participants, means of their identification and plan of the actual session.

The objectives of this stage were also included in the moderation guide for the focus group and consisted of the following elements:

1. Knowing the reaction of the participants to the results obtained in the research through the questionnaire;

2. Justification of the main results;
3. Determining the implications of these results;
4. Identifying the necessary measures to improve graduates' skills.

The focus group participants were selected from the following categories: 6 representative employers from the agri-food sector, 3 students selected from the interview stage belonging to three relevant faculties and interested in the subject of this research and 3 professors who teach subjects specific to the agri-food field. The organizers of the session were 3 members of the research team. One member had the role of moderator, one was responsible for recording information (logistics insurance) and one observed the participants' behavior.

Interaction with the participants took place online after previously informing them of the time, duration, objectives and way of working.

Data collection was performed in a focus group session that took place in the first quarter of 2023 and lasted three hours, with two 10 min breaks based on the moderation guide. The breaks included clarifications on how to interact during the sessions and question orientations structured around the four objectives of the study.

Participants shared opinions on these objectives and these were scored synthetically. After the first hour of discussion, lists of opinions were drawn up for each research objective (4 lists of opinions of the 12 people) and the participants were then asked to choose the components with which they agreed the most. They were given a limited number of 3 choices per list [63,64].

Data analysis began immediately after the debate ended. The research team led a short debate during which the results were centralized, correlated with the observations and the final results were determined.

The collected data were structured and centralized in order to draw up the investigation report. The analysis of the results obtained within the qualitative section on the study included a correlation between the objectives of the qualitative research and the profiles of the subjects. The subjects' common opinions and perceptions were highlighted, and the structure of their characteristics was recorded. Some subjects' discordant opinions were also highlighted in correlation with their profiles. In the former case, opinions in common were coded, identified and counted in order to establish their relationships with the subjects' characteristics. These were interpreted independently to allow us to outline the general opinion of the subjects and determine their general profile.

Dissenting opinions were identified by removing premiums and assessing relevance or importance to the research objectives. These results were interpreted separately, raising concerns about new approaches or issues requiring further debate.

### 2.2. Quantitative Research on Skills Relevant to Employment and Entrepreneurship in the Agri-Food Sector

The quantitative section of this study began with the establishment of research objectives for the questionnaire. The questionnaire was intended to determine the perception of relevant labor market actors regarding the skills needed by higher education graduates for employment or entrepreneurship in the agri-food sector. Its objectives consisted of the following elements: objective 1: quantifying the relative importance of higher education graduates' skills for employment in the agri-food sector; objective 2: quantifying the relative importance of higher education graduates' skills for starting and developing a business in the agri-food sector.

The targeted indicators were as follows: 1. the individual level of importance of each skill in relation to all others (score 1–100); 2. the average level of importance by categories of competencies and subjects (score 1–100); 3. the Pearson coefficient of multiple correlation between the level of importance of the competencies and the main characteristics of the subjects.

In this stage of the research, a questionnaire was used because it was necessary to collect a large amount of detailed information from a large number of respondents.

Questionnaires also offer the opportunity to analyze and interpret data in an efficient way [65].

The subjects of the questionnaire were IULS students, employers from the NE Region of Romania and IULS teachers. For the three categories of subjects, different samples were made with different objectives, different structures and different questionnaires. Their common element was the set of skills under research.

For each category of participants, the characteristics under analysis were as follows: for students, gender; for employers, the number of employees (0–10 employees, 11–50 employees and over 50 employees), the age of the organization (0–10 years, 11–20 years and over 20 years) and the type of organization (public or private); for teachers, experience in academic activity (0–10 years, 11–20 years and over 20 years). Employers were classified by business size as small, medium or large according to the SME Definition User Manual [66].

The characteristics according to which the subjects were structured were determined by the significant differences of opinion recorded during the interviews. Those features where polarized responses were obtained were considered relevant.

Although the main subjects of this study were employers, because they are the beneficiaries of the skills obtained by their future employees while at university, it was also necessary to know the perceptions of students in order to create the professional and entrepreneur profile towards which they tend. Also, teachers' perspectives on skills were investigated in order to know the foundation on which these educational efforts are built.

During the data collection stage, the human resources involved in administering the questionnaire were organized into teams of researchers specialized in the field of agri-food sciences and education.

The researched area was represented by employers from the NE Development Region of Romania who are active in the agri-food sector, IULS students, who mostly come from localities in the same region, and professors from three faculties within the university, i.e., Agriculture, Horticulture and Food and Animal Sciences.

The content of the questionnaire was created according to the results of the interviews. Skills were classified into 10 categories (communication, linguistic, mathematical, scientific, IT, learning, social, community, economic and cultural skills), each of which was divided into 3 subcategories (knowledge, skills and attitude). These were taken from the Council Recommendation of 22 May 2018 on key competencies for lifelong learning (2018/C 189/01) [67]. The group of competencies in the field of science, technology, engineering and mathematics was divided into mathematical competencies on the one hand and scientific competencies on the other hand because the interview subjects insisted on these two particular forms. For the same reason, the group of personal, social and learning skills was also divided into learning skills, on the one hand, and social skills, on the other. Practically, the definitions and the number of key skills did not changed, but two groups of skills were detailed (Table A1).

For a better understanding of these concepts by the subjects and to achieve an optimal size of the questionnaire, some definitions of these skills were synthesized. The questionnaire was created and administered online using the Google Forms platform [68].

The questionnaire intended for employers contained 67 questions, of which 30 questions (1–30) were related to objective 1 (quantifying the relative importance of higher education graduates' skills for employment in the agri-food sector). Another 30 questions (32–62) were related to objective 2 (quantifying the relative importance of higher education graduates' skills for starting and developing a business in the agri-food sector). A follow-up question was introduced at the end of each series (31, 63), asking respondents to add other skills they considered important. The last questions (63–67) referred to the subjects' profile: number of employees, age of the organization, turnover and type of organization (public or private).

In the first set of questions, regarding objective 1, employers were asked to assess the extent to which each competency is important for a potential job in the unit they manage.

In the second series of questions, regarding objective 2, employers were asked to assess the extent to which each competency is important for the activity they carry out as an entrepreneur (or member of the management team).

The questionnaire administered to students contained 66 questions, of which 30 questions (1–30) were related to objective 1 and another 30 questions (32–62) were related to objective 2. One supplementary question was introduced at the end of each series (31, 62) in which respondents were asked to add other competencies. The last questions (63–66) referred to the profile of the subjects: gender, year of study, faculty and specialization.

In the first set of questions, regarding objective 1, students were asked to rate the extent to which each competency is important for their potential employment after completing their undergraduate studies. In the second set of questions, regarding objective 2, students were asked to assess the extent to which each competency is important for the activity they could carry out if they had their own business.

The questionnaire intended for teachers contained 64 questions, of which 30 questions (1–30) were related to objective 1 and another 30 questions (32–61) were related to objective 2. One supplementary question was introduced at the end of each series (31, 62) in which respondents were asked to add other competencies. The last questions (63–64) referred to the profile of the subjects: teaching experience and teaching field.

In the first set of questions, regarding objective 1, teachers were asked to rate the extent to which each competency is important for a potential job. In the second series of questions, regarding objective 2, teachers were asked to assess the extent to which each individual competency is important for the activity carried out by an entrepreneur.

Questions that had the role of measuring the importance of skills were selection items with values on a 5-point Likert scale (1—very little important and 5—very important). These values measured the intensity of the subjects' perception of the importance of the analyzed skills [69]. The fill-in and profile questions were developed according to the type of short-text fill-in items.

The questionnaire was developed in the 2nd quarter of 2022 and was tested in the 3rd quarter on a small number of subjects: 5 employers, 22 students and 7 teachers. This stage revealed a deficiency regarding the way the competencies were assessed: most of the subjects rated all the competencies as important and very important (4–5 points). Consequently, the final form of the questionnaire included the following specification: "For each of these (skills), please select values from 1 to 5 according to the importance of the skill for entrepreneurs. Also, please adjust the ratings so that questions receive both 1 and 5 values".

The estimated duration of completing the questionnaire was approx. 20–30 min.

The questionnaire was administered in the 4th quarter of 2022 by email and messages on social networks. It was accompanied by the necessary explanations and a letter of accreditation from the Rector of IULS. The messages were accompanied, mostly in the case of employers, given their limited time resources, by telephone and direct conversations.

The subjects were assured of the anonymity of their responses and that they would be used exclusively for the purposes of the research. The individual data of individuals or organizations were stored in a protected channel with the guarantee of specialist researchers.

Data analysis started with the validation of the questionnaires. Data on the relative importance of the skills were converted into numerical values from 1–5 to 0–100 based on 100, and weighted coefficients (summing to 100) were calculated for better appreciation.

Data were processed using the software IBM SPSS Statistics 29 and Microsoft Office Pro plus 2021, with which size, correlation and comparative analyses were performed.

Multiple correlation analysis was carried out between the categories "skills for employment" and "skills for entrepreneurship" on the one hand and the profile characteristics of the subjects (number of employees, age of organization, teaching experience, etc.) on the other.

In the spirit of ethical standards specific to scientific research, the researchers made sure that their activity is relevant to society, that they do not duplicate other previously

conducted research and respect the principle of intellectual property and that the results of their research are made known through this article [70].

## 3. Results

*3.1. Opinions of Employers, Teachers and Students Regarding the Skills Needed in the Agri-Food Sector*

Semi-structured interviews were used to determine the opinions of teachers, students and employers regarding the skills needed in the agri-food sector. The reduction of the obtained data led to the development of a matrix and the elaboration of conclusions.

Interviewees generally felt that agri-food university graduates should develop lifelong learning, leadership and management skills, integrated knowledge of the agriculture and food industries and skills to identify career opportunities, as well as being prepared psychologically and financially to face the challenges of the field. These skills would enable them to adapt to changes in agriculture and build successful and sustainable careers.

The responses of students regarding the skills required of higher education graduates to find employment in the agri-food sector were dominated by the group of competencies aimed at skills (Table 1). Most of the students believed that graduates must have practical skills so that, at the time of employment, they can perform the tasks they receive from employers. Some of them believed that it is necessary to acquire specialized language and, in general, communication skills. When they were asked to list the most important skills, they specified the skills in this order: practical, technological, digital, motivation. Some students emphasized attitudes and behaviors, such as persuasion, analytical thinking, punctuality, attention to detail and a positive attitude. There were also views expressing a strong reluctance towards university education. The students who shared these opinions appreciated that "there is a big difference between theory and practice" and "practice requires something else than what is taught in college".

**Table 1.** Matrix of synthetic results obtained during the interviews.

| Objectives/Students Group | Students | Employers | Teachers |
|---|---|---|---|
| Skills for employment | technological, communication, linguistic, practical, adaptability | information technology, mathematics, new technologies, organization | scientific, lifelong learning, digital, linguistic |
| Priority skills | technological, digital practice, motivation | attitude, math, teamwork | scientific, lifelong learning, digital |
| Specific entrepreneurship skills vs. employment | finance, management, leadership, creativity, failure management | lifelong learning, teamwork, economics, stress management | scientific, economic, leadership |
| Behavior of subjects | insecurity, low motivation, reluctance | interest in the topic, mistrust of change | interest in the topic |

Most of the students thought that to start a business in the field, they need skills in the economic sphere (finance, management, leadership) but also of a psychological nature (creativity, failure management). They considered it necessary to have information about how to start a company, how to organize a possible team or how to make quality products. These aspects were also specified in a previous study, in which it was concluded that the performance of employees in a period of crisis can be influenced by competent crisis management and can increase support for teamwork [71].

However, a small part of the students saw entrepreneurship as a possible direction of professional development. Most students saw themselves after graduation in the position of employee and thought they could possibly start a business later. This phenomenon is observed less in students from agricultural specializations and more in those in the food industry. The exception is students who come from families that already own businesses.

Especially with regard to entrepreneurial skills, students had vague answers and showed insecurity. Even if a climate of trust was created at the time of the interview, the students answered rather formally and impersonally.

Employers indicated a multitude of skills needed for employment of graduates in the agri-food sector, such as lifelong learning, teamwork, economic skills and others. Some managers were deeply dissatisfied with the performance of graduates. One manager, the owner of a farm with a turnover of less than EUR 100 thousand, said that graduates are "completely unprepared" and that it would be necessary for them "to be taught to prepare the machines for agricultural work, to calculate the doses of inputs and know how to fight pests". This opinion was also presented by PwC (2019) as part of a study on the skills required in the Romanian food and beverage industry [42]. Most employers appreciated that the skills taught at universities should be tailored according to market needs. In contrast, when a manager from a multinational company was asked to rate the top three most important competencies needed for employment, he confidently stated that attitude was the top competency, followed by continuous learning and willingness to work in a team. He believed that employers have an obligation to provide their employees with current information and help them develop practical skills. This is enabled by an open attitude towards learning, collaboration and work among employees.

In order to start a business in the agri-food field, most managers believed that graduates need knowledge of lifelong learning, teamwork and stress management. Moreover, they need market, product, consumer and competition knowledge. Some subjects had a pessimistic attitude towards businesses created by recent graduates because the competitiveness of the field is constantly increasing and "most markets have been taken". They also recalled the entrepreneurial opportunities enjoyed by graduates from the post-communist generations when the transition to a market economy was made. In contrast, most managers believed that the current technological and economic development creates countless opportunities for starting and growing a business. This is why they believed that the ability to adapt to market conditions and take advantage of opportunities is necessary.

Most teachers' answers regarding the skills required from graduates to find employment in the agri-food sector highlighted the need for learning because any information can be useful at some point. They highlighted the multitude of directions and specializations that graduates' careers can focus on and that knowledge diversification may be more relevant than their specialization. The teachers believed that scientific knowledge, lifelong learning, digital and language skills can contribute to the professional success of graduates. These were also the priority competencies they specified. Digital skills were recognized as imperatives of our age [72,73].

In order to start a business in the field, the professors believed that graduates need at least as many scientific skills as managers in order to properly manage technical and human resources. They evaluated the acquisition of academic skills as a determining condition for success in business, although some of them admitted that in previous generations, some stood out as successful entrepreneurs while not having exceptional academic results. This is also the reason why they insisted on the need to improve economic skills. In addition, many believed that, especially in Romanian society, the specific profile of a leader is important in the conditions of the transition to a consolidated economy.

Following information processing from all three categories of subjects (students, employers and teachers), the purpose of the interviews was partially achieved. The information obtained did not lead to the concretization of a set of skills that would be the basis for the development of the questionnaire in the next stage. On the one hand, various skills were highlighted and these can be considered relevant for graduates' careers. Conversely, on the other hand, these skills cannot be structured into a logical system of skills. The competencies stated and often associated by the subjects belong to different logical groups, such as scientific knowledge and practical skills to perform technical activities or communication skills and attitude towards learning. Consequently, it was necessary to identify a system of competencies in which most of those highlighted by the subjects could be found.

Thus, the competency system adopted was devised according to the results of the interviews but using the internationally agreed upon structure specified in the Council Recommendation of 22 May 2018 on key competencies for lifelong learning (2018/C 189/01) [67]. Most of the skills specified by the interview respondents can be found in this classification. However, the need to classify skills into ten categories, instead of eight as specified in the document, was highlighted. These categories were communication, linguistic, mathematical, scientific, IT, learning, social, communication, economic and cultural skills. Each of these contains three subcategories (knowledge, skills and attitude). The definitions and number of key skills did not change, but two groups of skills were detailed. The group of competencies in the fields of science, technology, engineering and mathematics was divided into mathematical competencies on the one hand and scientific competencies on the other because the interview subjects insisted on these two particular forms. For the same reason, the group of personal, social and learning skills was also divided into learning skills on the one hand and social skills on the other. This approach is in agreement with You, M. and Ju, Y., who appreciate that a minimum package of skills in the field of agriculture has similarities with the eight key skills established by the Council of Europe [74].

*3.2. Perceptions of Relevant Labor Market Actors Regarding the Importance of Skills Needed by Higher Education Graduates for Employment or Entrepreneurial Activity in the Agri-Food Sector*

The administration of the questionnaire led to an average response rate of 80.4%. Employers answered 111 out of 127 administered questionnaires, students answered 288 out of 384 administered questionnaires and teachers answered 139 out of 158 administered questionnaires. Following their validation, a valid survey rate of 79.8% was found for the entire research sample.

Of the whole sample, the most important competency needed by higher education graduates in the agri-food field (Figure 1) proved to be communication, with a value of 90.5 points, followed by learning (74.3 points) and social skills (70.6 points). The least appreciated skills were cultural, linguistic and mathematical skills, with values of 17.5 points, 36.9 points and 43.8 points. Also, communication was rated as the most important skill to start a business (91.0 pts.), as in the case of employment-related skills, closely followed by economic skills (81.4 pts.) and learning skills (75.4 pts.). These results are consistent with another study, which considered crisis communication skills, leadership and decision-making to be important [19].

It is obvious that all the research subjects believe that, to find employment, higher education graduates need a knowledge of vocabulary, functional grammar and communication styles in different situations; the ability to communicate both orally and in writing in a variety of situations and to monitor and adapt one's own communication to the requirements of each situation; a positive and responsible attitude towards communication, and critical, constructive dialogue; and interest in interacting with others. Consequently, in an era of fluid communication unprecedented in the history of human civilization, employers, students and teachers in the agri-food field believe that the main skill needed is that of communication.

Also, learning skills are in increasingly demand. The required learning skills are as follows: knowledge and understanding of one's own preferred learning strategies, strengths and weaknesses; ability to seek available education, training and support opportunities; effective organization of autonomous and collaborative learning, career and work; persistence in learning; long-term focus and critical reflection on learning goals; motivation and confidence in the need for lifelong learning; and a problem-solving orientation in the face of various changes, obstacles and a variety of life contexts. Although the motivation of this study is one of improving the learning process in an institutional setting, all participants in this process appreciate this competency as a particularly important one. These results may raise questions about the satisfaction with educational services and highlight the importance of continuous learning in a spectacular dynamic of human intelligence.

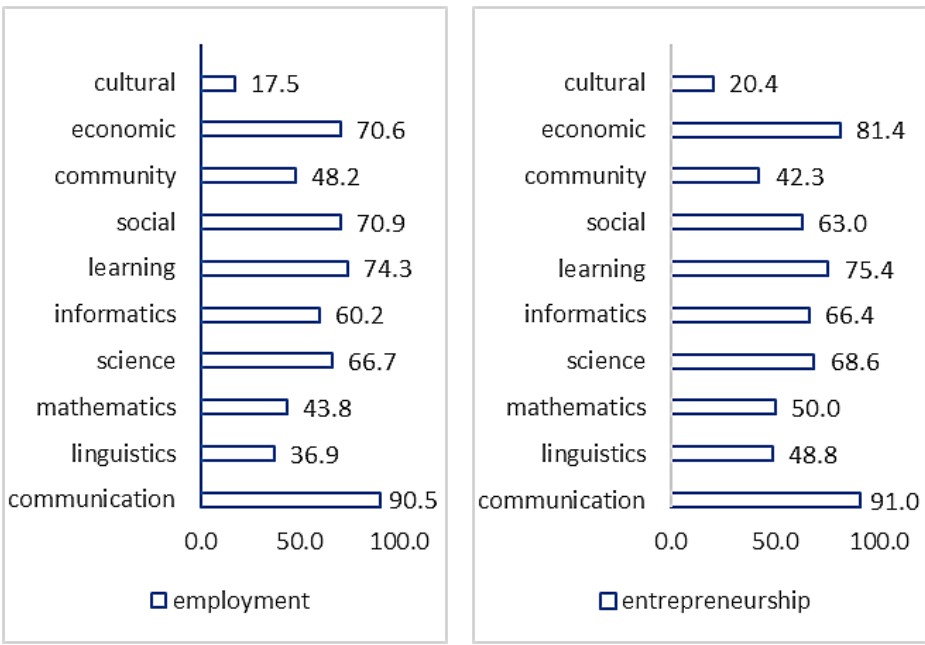

**Figure 1.** Cumulative general responses of employers, teachers and students regarding the skills needed for employment (**left**) and those needed for entrepreneurial activities (**right**).

Social competencies are largely required by the labor market in the agri-food sector. These include understanding how individuals can ensure optimal physical and mental health in accordance with gender equality and social and cultural non-discrimination; recognition of generally accepted rules, codes of conduct and manners in different societies and environments; communicating in a constructive, tolerant and empathetic way in different contexts; the ability to create trust; effective management of stress and frustration; the ability to distinguish between personal and professional areas; collaborative attitude, integrity, interest in society; and a respect and appreciation for the diversity of others and overcoming prejudices. The development of science and technology puts society at the center of the equation. Individuals are no longer self-sufficient entities due to the complexity of today's world. It is necessary to work in teams, to collaborate intra- and interdisciplinarily. Tolerance and recognition of and concern for one's own person together with others are now proving to be a sine qua non condition for developing a successful career.

In order to set up and develop a business, higher education graduates will first of all need the same skills of communication and continuous learning, but especially associated with economic skills. Future entrepreneurs need to understand the workings of the economy; the context, opportunities and challenges faced by the individual in their own work, in the work of their employer and other organizations; apply proactive management to their own work (identify opportunities, plan, organize, negotiate, manage, lead, delegate, analyze, communicate, inform and evaluate); and motivate themselves through initiative, proactivity, independence and innovation in the workplace. Economic skills have become much more comprehensive. They have become necessary tools for any facet of human activity, from optimizing costs to making relationships with oneself and one's family more efficient (see the expression "quality time"). Under these conditions, a relevant system of key competencies will be based on dynamic and elevated forms of economic competencies.

Employers believed that the most important competency needed by higher education graduates in the agri-food field is communication (Figure 2), with a value of 87.7 points, followed by social skills (85.1 points) and learning skills (82.5 points). The least important skills for employers were cultural, linguistic and community skills, with values of 14.9 points, 36.8 points and 44.7 points. Employers' perceptions did not change much

regardless of the number of employees under them. Employers with fewer than 10 employees (micro-enterprises) and those with more than 50 employees (medium and large enterprises) believed that graduates must first of all have communication skills (87.8 points and 88.7 points, respectively). Only those with 11–50 employees (small businesses) believed that the main skills for which they would hire higher education graduates would be communication and learning, to the same extent (88.6 pts). Neither did the age of the organization change the perspectives of employers. Those with under 10 years of experience believed that the most important competency is communication (88.5 pts.), but those with 10–20 years of experience believed that the main competency for which they would hire higher education graduates would be scientific knowledge (94.4 pts.) and those with over 20 years of experience believed that graduates must first of all have communication skills (88.6 pts.) and, to the same extent, learning skills (88.6 pts.). Managers of public organizations rated communication with 93.3 points and those of private ones with 88.3 points. These results did not converge to a general opinion that would indicate the importance of digital skills [21], probably because the subjects believed that digitization is accessible to them as long as they have real communication and learning skills.

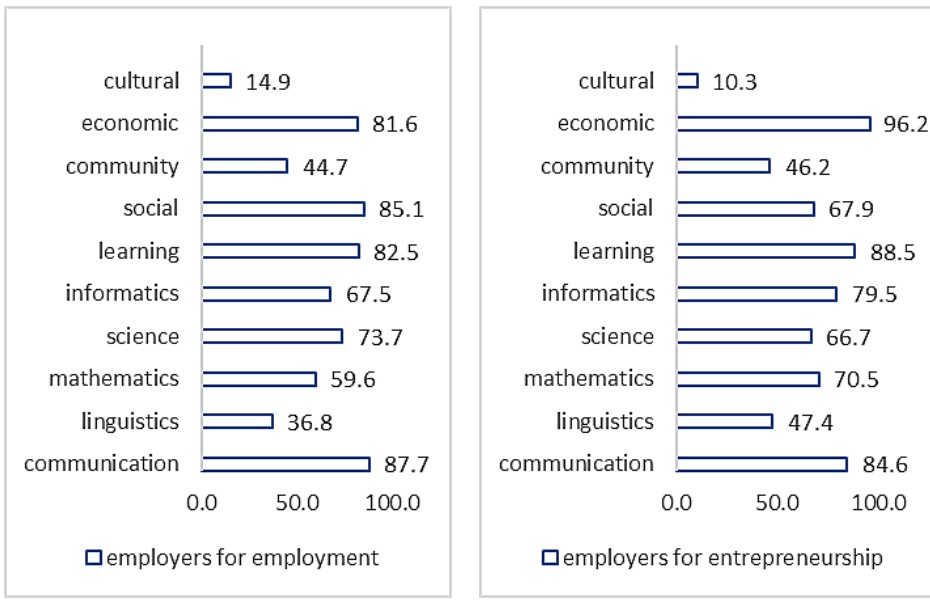

**Figure 2.** Employers' responses regarding the skills needed for employment (**left**) and those needed to carry out entrepreneurial activities (**right**).

Employers believed that, to find employment, graduates must have a positive and responsible attitude towards communication, the ability to conduct critical, constructive dialogue and an interest in interacting with others. Socially, they must demonstrate an attitude of collaboration, integrity and interest in society as well as value and respect the diversity of others and overcome prejudices. The learning competencies valued in graduates by employers are motivation, confidence and an awareness of the need for lifelong learning as well as problem-solving in the face of various changes, obstacles and a variety of life contexts.

The least important thing for a graduate's success as an employee, from the perspective of employers, is expressing creativity and a desire to cultivate esthetic abilities through artistic self-expression and participation in cultural life. It does not seem necessary for graduates to demonstrate an interest and curiosity in cultural diversity, foreign languages and intercultural communication. And it is not a priority for them to show solidarity and interest in solving problems that affect the local and wider community either.

The most important competency for starting a business in the field, as specified by the employers, was economic skills (96.2 pts.), followed by learning skills (88.5 pts.) and

communication skills (84.6 pts.). The least important skills for employers were cultural, community and linguistic ones, with values of 10.3 points, 46.2 points and 47.4 points. Regardless of the number of employees, the employers still put economic competency first (under 10 employees: 91.4 pts.; 11–50 employees: 92.5 pts.; greater than 50 employees: 92.8 pts.). The age of the organization only changed the intensity with which the need for economic competency was perceived (under 10 years: 91.1 points; 10–20 years: 91.8 points; more than 20 years: 92.5 points). Public organizations rated the importance of economic skills with 98.7 points and private ones with 91.0 points.

In order for graduates to run successful businesses or to achieve high-performance management in management teams, they first of all need to apply proactive management to their own work. This is associated with a positive attitude towards learning and a positive and responsible attitude towards communication as well as critical, constructive dialogue and an interest in interacting with others. Basic knowledge of local, national and European cultural heritage and their place in the world does not seem to be important for business development. Community skills, interest in and curiosity for cultural diversity, foreign languages and intercultural communication are also not found in the portfolio of a potential entrepreneur.

Students appreciated that the most important competency is communication skills, with a value of 96.0 points, followed by economic skills (74.6 points) and learning skills (73.2 points). The least important skills for students were cultural, mathematical and linguistic skills, with values of 30.8 points, 31.2 points and 31.9 points, respectively (Figure 3). Women thought that the most important competency for employment is communication (94.3 pts.), and men thought that the main competency for which they would hire higher education graduates would also be communication (94.3 pts.).

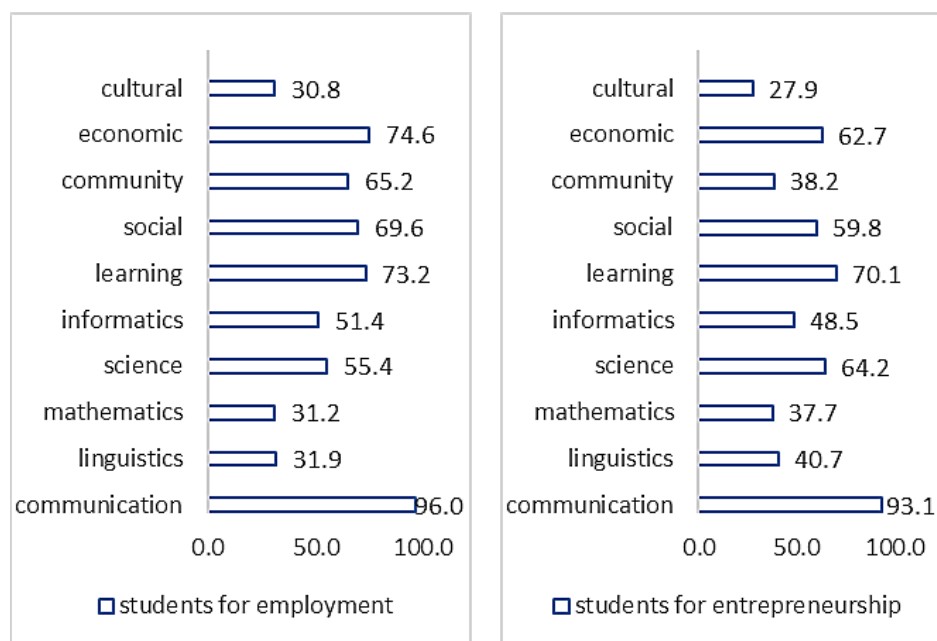

**Figure 3.** Students' answers regarding the skills needed for employment (**left**) and those needed to carry out entrepreneurial activities (**right**).

First of all, students thought that they need a knowledge of vocabulary, functional grammar and communication styles appropriate for different situations. Then, they believed that they must be able to identify opportunities and plan, organize, negotiate, manage, lead, delegate, analyze, communicate, inform and evaluate them. All this was associated with the need for motivation and confidence in lifelong learning and orientation towards problem-solving in the face of various changes, obstacles and a variety of

life contexts. Cultural knowledge and skills to apply basic mathematical principles and processes in different contexts, to reason mathematically, to communicate in a mathematical language and to use appropriate means are not important for employment. It is also of little importance to have an interest in and curiosity for cultural diversity, for foreign languages and for intercultural communication.

In order to set up a business in the agri-food sector, the most important value was, according to students' opinions, communication, with a value of 93.1 points, followed by learning (70.1 points) and scientific skills (64.2 points). The least important skills for entrepreneurship from the students' perspective were cultural, mathematical and community skills, with values of 27.9 points, 37.7 points and 38.2 points, respectively. Gender did not change the rank of skills but only the intensity of perception, with women valuing communication at 93.9 points and men at 93.7.

In order to operate a business, students believed that they must have, in addition to communication knowledge, other communication skills such as the ability to communicate both orally and in writing in a variety of communication situations and to monitor and adapt their own communication to the requirements of each situation. Scientific competencies were appreciated by students, especially in terms of knowledge of fundamental scientific concepts, principles, methods, technologies, products and technological processes; all this at a general level and at a domain-specific level. All this is possible if they have the motivation and determination through initiative, proactivity, independence and orientation towards innovation.

Teachers believed that the most important competency for employment is also communication, with a value of 87.7 points, followed by scientific knowledge (71.0 points) and learning skills (67.3 points). The least important skills according to the teachers were cultural, community and mathematical skills, with values of 6.8 points, 34.6 points and 40.7 points, respectively (Figure 4). The 2020 report by the Romanian Flour, Baking and Beef Industry Association (ROMPAN) shows that there is a growing need for technical and managerial skills in the Romanian food industry [41].

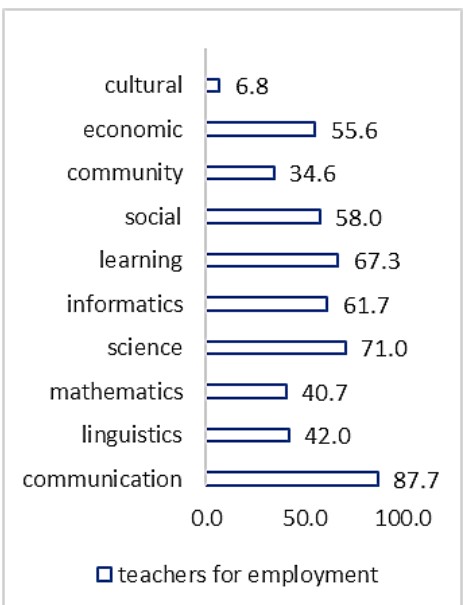 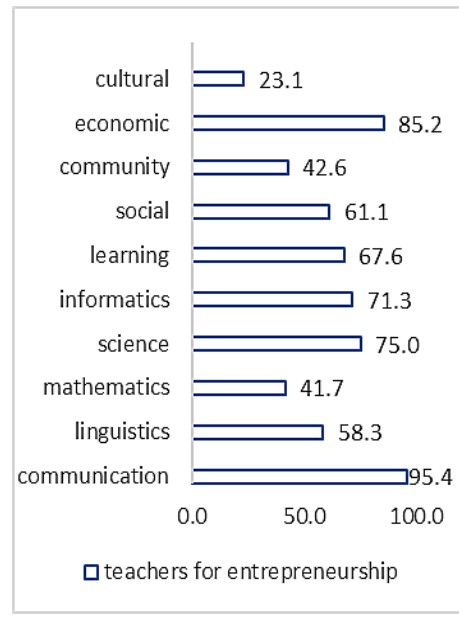

**Figure 4.** Teachers' answers regarding the skills needed for employment (**left**) and those needed to carry out entrepreneurial activities (**right**).

Experience only influenced the level of importance teachers gave to communication. Those with fewer than 10 years of experience gave it 92.1 points, those with 11–20 years of experience gave it 92.3 points and those with more than 20 years of experience gave it 92.5 points.

Generally, teachers believed that graduates who intend to work need the skills to communicate both orally and in writing in a variety of communication situations and to monitor and adapt their own communication to the demands of each situation. In relation to the development of scientific knowledge, graduates must have a critical attitude, curiosity for and interest in ethical issues and a respect for safety and sustainability, especially in relation to scientific and technological progress in relation to themselves and others.

For graduates looking to manage their own business, teachers believed the most important competency is communication (95.4 pts.), followed by economic skills (85.2 pts.) and scientific skills (75.0 pts.). Less important for teachers were cultural, mathematical and community skills, with values of 23.1 points, 41.7 points and 42.6 points, respectively. As in the case of skills needed for employment, experience only influenced the level of importance the respondents gave to communication skills. Teachers with fewer than 10 years of experience gave them 90.7 points, those with 11–20 years gave them 89.7 points and those with more than 20 years of experience gave them 89.6 points.

Teachers prioritized two components of communication, namely communication knowledge and skills, as particularly important in the context of running a business. These include knowledge of vocabulary, functional grammar and communication styles applicable in different situations as well as skills to communicate both orally and in writing in a variety of communication situations and to monitor and adapt one's own communication to the demands of each situation. In economic terms, communication skills are accompanied by motivation and determination through initiative, proactivity, independence and innovation at work. From a scientific point of view, teachers insisted on the knowledge of fundamental scientific concepts, principles, methods, technologies, products and technological processes; all this at a general level and at a domain-specific level.

Regarding the relationship between attitudes, skills and competencies (Figure 5), all subjects believed that attitude is the most important (62.0 pts.). However, the difference between attitude and knowledge, which was the least highly ranked category, is not large (56.2 points). Employers considered attitude to be the most important group of skills for employment, with a value of 66.1 points, and skills to be the most important group of skills for entrepreneurship, with 71.5 points.

The most important attitudes required for employment are the following: a positive and responsible attitude towards communication and critical, constructive dialogue and an interest in interacting with others; expressing respect for the truth and a willingness to look for reasons and evaluate their validity; motivation for, confidence in and awareness of the need for lifelong learning; problem-solving in the context of various changes, obstacles and a variety of life situations.

The most important sets of attitudes required for an entrepreneur are motivation and belief in the need for lifelong learning; problem-solving in the context of various changes, obstacles and a variety of life contexts; and motivation and determination through initiative, proactivity, independence and innovation in the workplace.

Students considered attitude to be the most important group of skills for both employment, with a value of 63.0 points, and for entrepreneurship, with 60.1 points.

They believed that to find employment, it is most important to exhibit a positive and responsible attitude towards communication and critical, constructive dialogue as well as an interest in interacting with others. The same competency is most important for starting and developing a business, according to the students.

Teachers also considered attitude to be the most important group of skills for both employment, with a value of 56.7 points, and for entrepreneurship, with 66.9 points.

Teachers' responses primarily recommended that graduates express a respect for truth and a willingness to look for reasons and evaluate their validity in order to find employment. For entrepreneurship, teachers recommended motivation and determination through initiative, proactivity, independence and innovation.

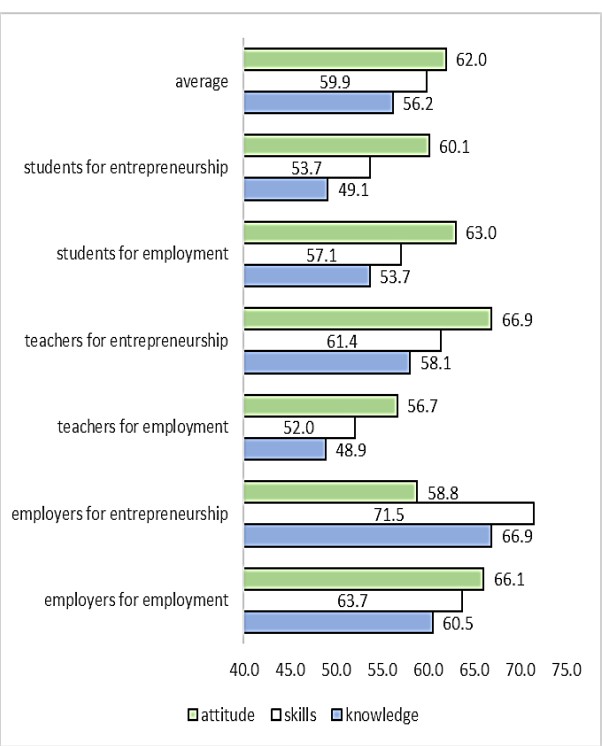

**Figure 5.** Subjects' responses regarding the groups of skills required for employment and those required for carrying out entrepreneurial activities.

Multiple correlation analysis indicated two correlations. There was a moderate inverse correlation (Pearson −0.60 assured for a level of 0.01, two-tailed) between the age of organization and the perceived importance of a positive and responsible attitude towards communication, critical, constructive dialogue and an interest in interacting with others. This suggests a lower interest of employers with consolidated time in management in attitudinal communication competency.

The number of employees correlated weakly and negatively (Pearson −0.49 secured for a level of 0.01, two-tailed) with attitudinal economic competency, which assumes that graduates, to succeed in employment, must be motivated and determined through initiative, proactivity, independence and innovation on the job. This correlation is justified by the fact that employers with a small number of employees have considerable difficulties regarding the management of human resources because the mobility of young employees is very high. Young employees gain experience in economic units that are less professionally attractive but more open to employment and then migrate to employers that provide higher job satisfaction. No statistically significant correlations were obtained for teachers and students.

## 4. Discussion

The focus group session took place with the participation of six employers (out of nine invited) representative of the agri-food sector, three students, three professors who teach subjects specific to the agri-food field and three members of the research team. The objectives of the focus group session were achieved by synthesizing the opinions presented by the participants.

Generally, the participants in this session had a positive attitude towards the study results, were interested in identifying the underlying causes of graduate success and were cooperative in building measures to improve the skills system specific to graduates in the agri-food sector.

### 4.1. Reactions of Participants to Results Obtained in the Questionnaire

Most of the participants agreed with the research results. Some of the entrepreneurs also reinforced the fact that, indeed, communication, social aspects and continuous learning represent the main vectors of success in terms of career development (Table 2).

**Table 2.** Matrix of synthetic results obtained in the focus group.

| Objectives/Students Group | Students | Employers | Teachers |
| --- | --- | --- | --- |
| Participants' reaction to the results | Agreement | Slight disagreement | Agreement |
| Justification of the main results | The complexity of today's communication, the unprecedented dynamics of technical progress | Not enough practice, school focused on knowledge and less on skills and attitude | The complexity of today's communication |
| Determining the implications | Poorly motivated employees, underperforming entrepreneurs | General feeling of insecurity, reactive attitude | Deficient economic education in the family and in society |
| Necessary measures | Information on priority skills | Developing relationships between students and entrepreneurs | |
| Behavior of participants | Concerned, cooperative | Slightly oppositional | Concerned, cooperative |

Some teachers appreciated that among all the competencies provided in the questionnaire, cultural competencies have the lowest values because, in an employee's daily interactions, responsibilities and usual activities, cultural competencies are needed less often. In contrast, in the activity of an entrepreneur, there is a higher need for cultural skills in situations where the entrepreneur has work meetings or negotiations with people from other localities, regions, countries, etc.

Participants noted a discordance in the teachers' responses. Economic knowledge seems to be more important for students for the purposes of employment than entrepreneurship. This could be attributed to a lack of information about what entrepreneurship means and the skills needed to develop a successful business. Navigating a changing world, networking and strategic development are also relevant in European, Middle Eastern and African research [75]. The results obtained are consistent with those of other studies, which show that there is a growing need for technical and managerial skills in the Romanian food industry [41].

The importance given by employers to entrepreneurial skills over knowledge was also noted, although both students and teachers put more emphasis on knowledge. And some students said that in college, they are evaluated on what they know, not on skills or attitude. They believed that there is a discrepancy between the theoretical knowledge they obtain in college and the practical applications expected of them as employees or entrepreneurs. They also thought that they carry out too few practical activities and these are not sufficiently well organized. Some professors stated that the university carries out practical activities in teaching farms, in micro production workshops, laboratories and at practice partners. However, they are not always relevant to the professional development of students due to some objective aspects. Entrepreneurs recognize that they do not always have the availability to offer internships for students. A manager at a company with fewer than 10 employees said he "doesn't have time to deal with students in practice and can't block an engineer" to do it. On the other hand, some managers from companies with more than 50 employees said that they are interested in accepting students for internships because this way they can ensure their human resources needs in the future. They believe that their companies' efforts are rewarded by improving human capital in the medium and long term.

### 4.2. Justification of the Main Results

Most of the participants appreciated that, first of all, communication is considered a very important competency because communication channels have become more and more complex, the volume of information is very large, and a superior ability to use information channels and select the volume of information is required. Communication skills in crisis situations, leadership and decision-making are specified in an article by Confort, L.K. et al. [19]. Some of the teachers believed that the need for communication has been heightened even more due to the period of the COVID-19 pandemic, during which some courses were taught online. A positive effect of this period was also the fact that they developed their digital skills. The importance of digital skills was also recognized as a priority of the current era [72,73]. Also, results obtained in another study that assessed the skills and knowledge needs of future professionals in the agri-food and forestry sectors emphasized the importance of improving digital skills, management, food safety, quality management, efficient use of resources, organization and planning [76].

Students' perceived insecurity in terms of communication could be one of the reasons why they believe that they need time and effort to develop specific skills to cope in various situations. During the hiring process, the difficulty of communicating during interviews could be the first moment when graduates acutely perceive the need for this skill. The absence of practical experience in real communication situations, critical self-evaluation, the constant tendency to compare themselves with others and the exacting standards imposed could be some of the reasons behind the students' answers.

The unprecedented dynamics of most fields of knowledge require a constant self-education effort based on new knowledge, current skills and a positive attitude towards progress. Some entrepreneurial participants believed that this development is dangerous and induces increased pressure on human resources with a higher education level.

The interest in communication, economic and learning skills is explained by the fact that the respondents to the questionnaire, especially the categories of teachers and employers, who already had professional experience either as employees or as entrepreneurs, found that in most situations, communication simplifies work tasks, negotiations, dialogue, collaboration, etc. A lack of effective communication can make it harder and longer to understand or solve a task.

Economic skills were unanimously appreciated as particularly important given that Romanian society does not have a traditional economic culture. Families do not have the capacity to educate their children from an economic perspective. Even entrepreneurship is seen as a very risky career alternative. Parents educate their children especially for employment and very rarely consider the possibility of them opening their own business. Here, the students emphasized the fact that they feel very insecure in the area of economic knowledge because in pre-university school, they took very few economic courses which were generally theoretical. During college, this knowledge was strengthened in relation to practice, but especially in economic specializations. Other students pointed out that most of their peers only think about getting a job after college, and very few of them consider the entrepreneurial alternative. Economics is often perceived by students as an extremely complex field, with the impression that they lack the mathematical skills or other fundamental disciplines needed to understand certain economic phenomena.

### 4.3. Determining the Implications of These Results

Employers believed that the shift from a knowledge-based skills system specific to the end of the last century to a multipurpose skills system has negative effects on all categories of entities concerned. Teachers stated they have to make significant efforts to match university curricula to market demand, while students are disoriented and disinterested in performance. An employer from a private organization with over 50 employees and over 20 years of experience stated that he prefers to hire students with minimum knowledge and skills but who "have a good attitude and we will teach him the rest". Another study with



the same objective pointed to the need for an adaptable workforce with interdisciplinary skills, especially for lower and middle management roles [77].

Students overwhelmingly said that there are not enough practical opportunities for skill development. Some employers contested this situation because they received students for internship who only visited the companies and had no apparent professional interests.

A professor specialized in the field of economics stated that obtaining the cooperation of students in specific classes requires great efforts because students believe that "only business people should know economics". Economic skills, although assessed as very important in this study, do not seem to represent actual concerns of students in the education sector.

Employers said they have noticed a reluctance among young employees towards improving their knowledge and skills rigorously. According to the employers, graduates generally do not have a positive attitude towards learning. Moreover, they lack a positive attitude in general: towards work, the company's objectives, teamwork. An employer presented an example in which a young employee "is here (at work) but as if he is not here". These situations probably also occur due to some discordances highlighted by Comfort, L. and collaborators: employers want graduates to come prepared with transferable skills, but many graduated have not demonstrated these skills [19].

However, most of the participants positively appreciated the fact that the issue of multiple skills is known to most graduates and educational institutions are taking significant steps in this regard.

*4.4. Identifying the Necessary Measures to Improve Graduates' Skills*

According to the employers, a logical and necessary measure following the identification of these implications is to publicize the results of this research. The dissemination of this information must be made among university staff, employers, entrepreneurs and certainly not least among students and their families. The results of this study, supplemented by others, can serve as a basis for the dynamic development of educational programs and for the adaptation of curricula so that more emphasis is put on the development and improvement of students' skills.

A second direction of action consists in strengthening current partnerships and creating new partnerships between universities, the economic environment and public entities and research institutes that offer employment. Recognizing the common interest in developing the skills of graduates is a first step. Other studies reinforce the conclusion that in today's social, economic, natural and geopolitical context, there is an urgent need for research that looks at professional and entrepreneurial skills to address the challenges and opportunities of the rapidly changing global economy [8]. Students can achieve that by working with employers to build a solid foundation for a successful career and have a positive impact on organizations.

The teachers informed the participants that universities in Romania have committees whose purpose is to adapt educational offers to the demands of the labor market. The results of these committees could be more visible to allow the teaching staff to dynamically adapt their teaching programs.

The results of the study, both of a qualitative and quantitative nature, offer a specific perspective on the skills system needed by higher education graduates in the agri-food field. Graduates from this sector of activity need pragmatic skills that are useful for increasing performance in terms of both employment and entrepreneurship. Of course, students, graduates and young employees are not indifferent to cultural values, community values or linguistic tools, but they understand the importance of focusing their efforts on indispensable career skills in the current context of global evolution.

These results outline the direction in which it is necessary to restructure training efforts at universities, i.e., towards an increased involvement of stakeholders within public and private organizations and, in general, the collaboration of Romanian society in improving the academic performance specific to the new generations. The stakes of these efforts are

overwhelming because the young people studying at top universities now are the ones who will produce tasty, healthy, sustainable and fair food for all of humanity tomorrow.

*4.5. Research Limitations and Recommendations for Future Research*

The first limitation that was identified in this study consisted in the difficulty with which the dissociation between the competencies that participants consider important because they lack them and those they consider important because they are used specifically at work or in entrepreneurship was achieved. Future research could use interdisciplinary teams to enhance and inform the psychological aspects of this inquiry.

The second limitation was the low availability of employers to participate in the research for obvious reasons related to the economic nature of their activities. However, a better promotion of the research objectives could, in subsequent research, ensure they are more fairly represented. The fact that the education of future employees and entrepreneurs in the agri-food field is not a priority for employers compared to their daily activities suggests a reserved attitude towards the results of the university education system. However, this attitude changes over time and future research will benefit from this dynamic.

The extent and importance of the educational phenomenon mean that an integrated perspective cannot be obtained by a single project. Instead, by identifying the necessary resources and by creating intra- and interdisciplinary teams, the best ways to improve the performance of higher education graduates in the agri-food sector can be identified.

## 5. Conclusions

In the first part of this research study, various skills relevant to graduates' careers were highlighted, but it was necessary to identify a system of skills in which to find most of the skills highlighted by the subjects. Thus, an internationally agreed upon structure was adopted, as specified in the EU Council's recommendations. The structure consisted of eight competencies that were easily adapted to this field of research.

At the level of the entire sample, the most important competencies needed by higher education graduates in the agri-food field were communication, learning and social competencies, with values of 90.5 pts., 74.3 pts. and 70.6 pts., respectively. The least appreciated skills were cultural, linguistic and mathematical skills, with values of 17.5 pts., 36.9 pts. and 43.8 pts. For entrepreneurial activities, the subjects believed communication to be the most important competency (91.0 points), as in the case of employment-related skills. It was closely followed by economic skills (81.4 points) and learning skills (75.4 points).

Employers considered the most important competency to be communication, with a value of 87.7 points, followed by social skills (85.1 points) and learning skills (82.5 points). The least important skills for employers were cultural, linguistic and community skills, with values of 14.9 points, 36.8 points and 44.7 points.

Students rated communication, economic and learning skills as the most important, with values of 96.0 points, 74.6 points and 73.2 points, respectively. The least important skills for students were cultural, mathematical and linguistic skills, with values of 30.8 points, 31.2 points and 31.9 points, respectively. For entrepreneurship, the students rated communication as the most important, with a value of 93.1 points, followed by learning (70.1 points) and scientific skills (64.2 points).

Teachers considered that the most important competency for employment is also communication, with a value of 87.7 points, followed by scientific knowledge (71.0 points) and learning skills (67.3 points). The least important competencies for teachers were cultural, community and mathematical skills, with values of 6.8 pts., 34.6 pts. and 40.7 pts., respectively. For graduates to manage their own business, teachers believed that the most important competency is communication (95.4 points), followed by economic skills (85.2 points) and scientific skills (75.0 points).

Regarding the relationship between attitudes, skills and competencies, all subjects believed that attitude is the most important (62.0 pts.).

Students' perceived insecurity in communication could be one of the reasons why they believe that they need time and effort to achieve specific skills to cope in various situations.

On the other hand, the unprecedented dynamics of most fields of knowledge require a constant self-education effort based on new knowledge, current skills and a positive attitude towards progress.

Students, entrepreneurs and teachers believed that additional efforts are needed to improve practical activities at universities in order to increase the relevance of university curricula to the skills required by the market. It is also necessary to strengthen the current partnerships and create new partnerships between universities, the economic environment and public employing entities and research entities.

**Author Contributions:** Conceptualization, D.B.; methodology, D.B. and A.-D.R.; software, D.D.; validation, A.S. and R.N.R.; formal analysis, R.-N.M.; investigation, C.C. and A.-D.R.; resources, R.N.R.; data curation, A.S.; writing—original draft preparation, D.B.; writing—review and editing, A.S. and A.-D.R. All authors have read and agreed to the published version of the manuscript.

**Funding:** This research was funded by the Romanian Ministry of Education, grant no. CNFIS-FDI-2022-0112/01.04.2022.

**Institutional Review Board Statement:** This research was carried out based on a mixed structure consisting of two qualitative research methods (interview and focus group) and a quantitative method, namely a questionnaire-based survey. All three methods involved interactions with, opinions of and answers from humans. The human subjects were informed at the beginning of the interview about the answers that would be requested and how they would be used. It was also explained to them that these would be published without naming their personal data. Accordingly, they were asked for their consent regarding the use of this information.

**Informed Consent Statement:** All subjects were informed at the beginning of the interview about the answers that would be requested and how they would be used. It was also explained to them that these would be published without naming their personal data. Accordingly, they were asked for their consent regarding the use of this information.

**Data Availability Statement:** A summary of the subjects' responses to the questionnaire is contained within this paper in Appendix A.

**Conflicts of Interest:** The authors declare no conflicts of interest.

# Appendix A

**Table A1.** Summary of the subjects' responses to the questionnaire.

| Key Competencies | Competency Groups | Employers: Important for Employment | | Students: Important for Employment | | Teachers: Important for Employment | | Average for Employment | Employers: Important for Entrepreneurship | | Students: Important for Entrepreneurship | | Teachers: Important for Entrepreneurship | | Average for Entrepreneurship |
|---|---|---|---|---|---|---|---|---|---|---|---|---|---|---|---|
| **communication** | knowledge | 84.2 | | 100.0 | | 87.0 | | | 80.8 | 84.6 | 100.0 | 93.1 | 100.0 | | |
| | skills | 84.2 | 87.7 | 91.3 | 96.0 | 100.0 | 87.7 | 90.5 | 84.6 | | 100.0 | | 100.0 | 95.4 | 91.0 |
| | attitude | 94.7 | | 96.7 | | 75.9 | | | 88.5 | | 79.4 | | 86.1 | | |
| **language** | knowledge | 44.7 | | 33.7 | | 38.9 | | | 53.8 | 47.4 | 63.2 | 40.7 | 63.9 | | |
| | skills | 44.7 | 36.8 | 37.0 | 31.9 | 55.6 | 42.0 | 36.9 | 80.8 | | 36.8 | | 80.6 | 58.3 | 48.8 |
| | attitude | 21.1 | | 25.0 | | 31.5 | | | 7.7 | | 22.1 | | 30.6 | | |
| **mathematics** | knowledge | 44.7 | | 19.6 | | 27.8 | | | 73.1 | 70.5 | 23.5 | 37.7 | 36.1 | | |
| | skills | 39.5 | 59.6 | 0.0 | 31.2 | 11.1 | 40.7 | 43.8 | 50.0 | | 14.7 | | 0.0 | 41.7 | 50.0 |
| | attitude | 94.7 | | 73.9 | | 83.3 | | | 88.5 | | 75.0 | | 88.9 | | |
| **science** | knowledge | 84.2 | | 58.7 | | 72.2 | | | 76.9 | 66.7 | 69.1 | 64.2 | 94.4 | | |
| | skills | 60.5 | 73.7 | 54.3 | 55.4 | 59.3 | 71.0 | 66.7 | 84.6 | | 67.6 | | 61.1 | 75.0 | 68.6 |
| | attitude | 76.3 | | 53.3 | | 81.5 | | | 38.5 | | 55.9 | | 69.4 | | |
| **information technology** | knowledge | 81.6 | | 60.9 | | 70.4 | | | 96.2 | 79.5 | 36.8 | 48.5 | 75.0 | | |
| | skills | 73.7 | 67.5 | 56.5 | 51.4 | 72.2 | 61.7 | 60.2 | 80.8 | | 54.4 | | 77.8 | 71.3 | 66.4 |
| | attitude | 47.4 | | 37.0 | | 42.6 | | | 61.5 | | 54.4 | | 61.1 | | |
| **learning** | knowledge | 73.7 | | 70.7 | | 70.4 | | | 88.5 | 88.5 | 72.1 | 70.1 | 52.8 | | |
| | skills | 78.9 | 82.5 | 69.6 | 73.2 | 64.8 | 67.3 | 74.3 | 84.6 | | 60.3 | | 75.0 | 67.6 | 75.4 |
| | attitude | 94.7 | | 79.3 | | 66.7 | | | 92.3 | | 77.9 | | 75.0 | | |
| **social** | knowledge | 63.2 | | 57.6 | | 48.1 | | | 42.3 | 67.9 | 48.5 | 59.8 | 38.9 | | |
| | skills | 100.0 | 85.1 | 70.7 | 69.6 | 61.1 | 58.0 | 70.9 | 88.5 | | 57.4 | | 61.1 | 61.1 | 63.0 |
| | attitude | 92.1 | | 80.4 | | 64.8 | | | 73.1 | | 73.5 | | 83.3 | | |
| **community** | knowledge | 50.0 | | 54.3 | | 22.2 | | | 61.5 | 46.2 | 23.5 | 38.2 | 36.1 | | |
| | skills | 28.9 | 44.7 | 57.6 | 65.2 | 24.1 | 34.6 | 48.2 | 34.6 | | 33.8 | | 33.3 | 42.6 | 42.3 |
| | attitude | 55.3 | | 83.7 | | 57.4 | | | 42.3 | | 57.4 | | 58.3 | | |

**Table A1.** *Cont.*

| Key Competencies | Competency Groups | Employers: Important for Employment | | Students: Important for Employment | | Teachers: Important for Employment | | Average for Employment | Employers: Important for Entrepreneurship | | Students: Important for Entrepreneurship | | Teachers: Important for Entrepreneurship | | Average for Entrepreneurship |
|---|---|---|---|---|---|---|---|---|---|---|---|---|---|---|---|
| **economics** | knowledge | 73.7 | | 70.7 | | 48.1 | | | 96.2 | 96.2 | 54.4 | 62.7 | 72.2 | | |
| | skills | 86.8 | 81.6 | 80.4 | 74.6 | 55.6 | 55.6 | 70.6 | 100.0 | | 64.7 | | 88.9 | 85.2 | 81.4 |
| | attitude | 84.2 | | 72.8 | | 63.0 | | | 92.3 | | 69.1 | | 94.4 | | |
| **culture** | knowledge | 5.3 | | 10.9 | | 3.7 | | | 0.0 | 10.3 | 0.0 | 27.9 | 11.1 | | |
| | skills | 39.5 | 14.9 | 53.3 | 30.8 | 16.7 | 6.8 | 17.5 | 26.9 | | 47.1 | | 36.1 | 23.1 | 20.4 |
| | attitude | 0.0 | | 28.3 | | 0.0 | | | 3.8 | | 36.8 | | 22.2 | | |

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
