# Peer review of "Relevant Skills for Employment and Entrepreneurship in the Agri-Food Sector"

_sustainability, doi:10.3390/su16020787_

Round 1

Reviewer 1 Report

Comments and Suggestions for Authors

ABSTRACT

The research topic is clear, well structure .

The topic is threat in correct form.

In general, the article is interesting and works in depth, I consider that the topic is in line with the journal’s research objectives.

INTRODUCTION:

The study objective identified.  In the introduction is well structured. Data and references are well. The objective is clear, and focused in agri-food sector.

The objectives of the research are clear.

 MATERIALS AND METHODS

The preparation of the research are good, the mix improved the article. All the methods are clearly exposed.

The interview are well structured, the groups correctly selected. The profile of the participants are well-structured in groups

The questionnaire is extense but structured, better for the analysis of the data. Good election realize three types of them

Good idea focused in a small group for exchange opinion

The structured is well defined, and it has enough information.

RESULTS:

The tables are well structured , clear and useful data

It is normal the vision of the students because of the economic situation

The explanation is well structured, and gives a global vision of the research

Figure 1-4 are clear, and the explanation after each  figure give us extra information

Figure 5 and the explanation complete the global vision

A possible line of research is observed. Well done

Well structured

DISCUSSION:

Clear and well structured., is an extra point  the different parts

CONCLUSION and future research:

An interesting line of research is observed, it could be go ahead in this research in local and international studies. Its shown relevant data from the groups and well structured.

Author Response

Thank you for your thoughtful and thorough feedback on our manuscript "Relevant skills for employment and entrepreneurship in the Agri-food sector". We acknowledge the time and effort required to review research manuscripts and greatly appreciate your dedication to academia and your contributions to the advancement of research.

Reviewer 2 Report

Comments and Suggestions for Authors

The article deals with employability skills in the agri-food sector.

In general, it is observed that the article complies with the proper principles of a scientific research and no serious errors are observed.

However, some advice that should be taken into account by the authors of the article is commented on.

It is noted that the article should be better structured in order to make it easier to understand and read.

To begin with, the introduction is considered too long. It is recommended to reduce it and that it serves to introduce the subject to be treated, commenting at the end of it the objectives of the study. The rest of the theoretical information in the introduction should be added as a new section of the theoretical framework.

Subsequently, in the methodology section, it is also recommended to order and structure the content. If there are two types of research (qualitative and quantitative), I would add two different subsections under methodology, one for qualitative and one for quantitative. For each of these two sections I would comment on the three fundamental aspects of any methodology: research design, data collection and data analysis.

Data analysis in qualitative research is discussed in a somewhat short form. It does not comment on how the categorization of the content has been carried out; in my opinion, the analysis that has been carried out needs to be better explained and argued. I understand that the analysis was performed manually, for future research it is recommended to use qualitative analysis tools such as Nvivo or Atlas.

The results, discussion and conclusions are well done.

Author Response

We appreciate the time and effort you have dedicated to providing feedback on our manuscript "Relevant skills for employment and entrepreneurship in the Agri-food sector". Thank you for your insightful comments, valuable improvements to our work, and the opportunity to increase our team's research performance.

We have followed your suggestions and made changes to the manuscript. These are highlighted within the manuscript.

Reviewer 3 Report

Comments and Suggestions for Authors

The paper refers to skills required by employees and entrepreneurs in the agri-food sector in Romania. The sample of  sample of 111 employers, 288 students and 139 teachers 13 from the North-East Development Region of Romania was used. It is questionable that teachers and students predominate in the sample, so the sample should be constructed in such a way as to ask mainly entrepreneurs. Students comment more on their ideas about success factors in the agri-food sector than on what these factors actually are. The Authors of the paper are aware of this, since they wrote that the limitation of presented research was the low availability of employers to participate in research for obvious reasons related to the economic nature of the activity they handle  (p. 23). The analysis of factors influencing modern food agriculture (especially in Romania) is done correctly. The overall quality of the paper should be assessed as average.

Comments on the Quality of English Language

The paper refers to skills required by employees and entrepreneurs in the agri-food sector in Romania. The sample of  sample of 111 employers, 288 students and 139 teachers 13 from the North-East Development Region of Romania was used. It is questionable that teachers and students predominate in the sample, so the sample should be constructed in such a way as to ask mainly entrepreneurs. Students comment more on their ideas about success factors in the agri-food sector than on what these factors actually are. The Authors of the paper are aware of this, since they wrote that the limitation of presented research was the low availability of employers to participate in research for obvious reasons related to the economic nature of the activity they handle  (p. 23). The analysis of factors influencing modern food agriculture (especially in Romania) is done correctly. The overall quality of the paper should be assessed as average.

Author Response

Dear Reviewer,

We appreciate the time and effort you have dedicated to providing feedback on our manuscript "Relevant skills for employment and entrepreneurship in the Agri-food sector". Thank you for your relevant comments, valuable improvements to our work, and the opportunity to increase our team's research performance.

We have followed your suggestions and made changes to the manuscript.

Reviewer 4 Report

Comments and Suggestions for Authors

1-In line 40 it says “ The agrifood sector is an essential component of many national and regional economies, providing food and jobs for millions of people around the world. In this context, it would be good to indicate data or GDP that reflects the importance  that is being talked about

2-When it refers to  as the percentage of highly qualified staff is low and the transfer of scientific knowledge and innovation to industry is limited, What percentage are you talking about and what are you comparing it to? Specify more

3-Materials and Methods. From my point of view I don't think it is necessary to explain the purpose of SPSS software, it would be enough to indicate that it was used, and the version used and the same for Microsoft Excel

4-The discussion should be revised.  It should include previous studies that support or contradict the results found.

Author Response

Dear Reviewer:

Thank you for your letter and the reviewers' comments on our manuscript "Relevant skills for employment and entrepreneurship in the Agri-food sector" (ID: sustainability-2797645). These comments are all valuable and very useful for revising and improving the paper, as well as important guiding significance for our research. We have carefully studied the comments and made additions accordingly, additions which we hope will be accepted. 
